# Realistic mossy fiber input patterns to unipolar brush cells evoke a continuum of temporal responses comprised of components mediated by different glutamate receptors

Vincent Huson, Wade G Regehr*

Department of Neurobiology, Harvard Medical School, Boston, United States

## eLife Assessment

This **valuable** study describes how trains of mossy fiber stimulation control cerebellar unipolar brush cell discharges. The dissection of the contributions of relevant glutamate receptors to these transformations is **convincing**. Overall, the study broadens our understanding of temporal processing in the cerebellar cortex.

**\*For correspondence:**
wade_regehr@hms.harvard.edu

**Abstract** Unipolar brush cells (UBCs) are excitatory interneurons in the cerebellar cortex that receive mossy fiber (MF) inputs and excite granule cells. The UBC population responds to brief burst activation of MFs with a continuum of temporal transformations, but it is not known how UBCs transform the diverse range of MF input patterns that occur in vivo. Here, we use cell-attached recordings from UBCs in acute cerebellar slices to examine responses to MF firing patterns that are based on in vivo recordings. We find that MFs evoke a continuum of responses in the UBC population, mediated by three different types of glutamate receptors that each convey a specialized component. AMPARs transmit timing information for single stimuli at up to 5 spikes/s, and for very brief bursts. A combination of mGluR2/3s (inhibitory) and mGluR1s (excitatory) mediates a continuum of delayed, and broadened responses to longer bursts, and to sustained high frequency activation. Variability in the mGluR2/3 component controls the time course of the onset of firing, and variability in the mGluR1 component controls the duration of prolonged firing. We conclude that the combination of glutamate receptor types allows each UBC to simultaneously convey different aspects of MF firing. These findings establish that UBCs are highly flexible circuit elements that provide diverse temporal transformations that are well suited to contribute to specialized processing in different regions of the cerebellar cortex.

## Introduction

Unipolar brush cells (UBCs) are a specialized type of excitatory interneuron in the cerebellum and cerebellar-like structures (*Altman and Bayer, 1977*; *Floris et al., 1994*; *Mugnaini and Floris, 1994*; *Rossi et al., 1995*; *Meek et al., 2008*). They receive a single mossy fiber (MF) input onto their large dendritic brush, and they in turn make their own mossy fiber boutons that make synapse mainly onto granule cells, but also onto other UBCs (*Rossi et al., 1995*; *Nunzi et al., 2001*; *Hariani et al., 2024*). UBCs transform MF firing patterns into prolonged increases or decreases in firing that lead to diverse granule cell firing patterns (*Rossi et al., 1995*; *Kinney et al., 1997*; *Diño et al., 2000*; *Mugnaini et al.,*

*2011*; *Locatelli et al., 2013*; *Kennedy et al., 2014*; *van Dorp and De Zeeuw, 2014*; *Borges-Merjane and Trussell, 2015*; *Zampini et al., 2016*; *Guo et al., 2021a*). In weakly electric fish, UBC firing provides a filtered signal that is suited to cancelling self-generated signals (*Kennedy et al., 2014*). In the rodent brain, recordings have been made from UBCs in vivo (*Simpson et al., 2005*; *Barmack and Yakhnitsa, 2008*; *Ruigrok et al., 2011*; *Hensbroek et al., 2015*; *Witter and De Zeeuw, 2015*), but in the absence of simultaneous recordings from connected MF-UBC pairs it is not known how UBCs transform MF firing patterns during behaviors.

Most of what is known regarding UBC temporal transformations is based on brain slice experiments in which MFs are activated with brief bursts (*Rossi et al., 1995*; *Mugnaini et al., 2011*; *Borges-Merjane and Trussell, 2015*; *Guo et al., 2021a*). UBCs responses range from fast 'ON' cells that fire for hundreds of milliseconds with virtually no delay, to intermediate UBCs that initially pause before increasing firing for seconds, to 'OFF' UBCs that suppress firing for up to two seconds (*Guo et al., 2021a*; *Huson et al., 2023*). Responses are mediated by a combination of different glutamate receptors. mGluR2/3s activate potassium channels to suppress firing (*Russo et al., 2008*; *Kim et al., 2012*; *Borges-Merjane and Trussell, 2015*), while a combination of AMPARs and mGluR1s mediate excitation (*Rossi et al., 1995*; *Kinney et al., 1997*; *Kim et al., 2012*; *Schwartz et al., 2012*; *Borges-Merjane and Trussell, 2015*; *Guo et al., 2021a*). Populations of UBCs exhibit a continuum of response durations and amplitudes, which is thought to reflect inverse expression profiles of the mGluR1 and mGluR2 signaling pathways (*Guo et al., 2021a*; *Kozareva et al., 2021*).

Little is known about how UBCs transform MF activity patterns during behavior. This issue is complicated by regional differences in MF firing properties (*Eccles et al., 1971*; *Lisberger and Fuchs, 1978*; *van Kan et al., 1993*; *Garwicz et al., 1998*; *Rancz et al., 2007*; *Arenz et al., 2008*; *Witter and De Zeeuw, 2015*). Even though UBCs are present throughout the cerebellar cortex, they are most abundant in regions involved in the control of eye position and in vestibular processing (*Floris et al., 1994*; *Diño et al., 1999*; *Takács et al., 1999*; *Englund et al., 2006*). MFs in these UBC-rich regions often fire in brief high-frequency bursts (*Eccles et al., 1971*; *van Kan et al., 1993*; *Garwicz et al., 1998*; *Rancz et al., 2007*) that in some cases represent saccadic eye movements (*Ohtsuka and Noda, 1992*). MFs can also fire continuously for extended periods and change their firing rates dynamically to represent motor signals, and encode velocity, position, and direction (*Lisberger and Fuchs, 1978*; *van Kan et al., 1993*; *Arenz et al., 2008*; *Barmack and Yakhnitsa, 2008*; *Lisberger, 2009*). Brain slice experiments suggest that UBCs can perform useful computations that transform MF inputs. UBC responses to sinusoidal modulation of MF firing rates have been previously examined (*Zampini et al., 2016*; *Balmer et al., 2021*; *Guo et al., 2021a*). It was found that 'OFF' UBCs provide a phase reversed response in which the UBC fires out of phase with the MF, while AMPAR responses provide a variety of phase transformations (*Zampini et al., 2016*). Such temporal transformations are thought to provide granule cell activity patterns that can be used in combination with plasticity mechanisms to modify the phase of the Purkinje cell output. However, contributions from mGluR1 were not assessed in that study. The mGluR1 component is prone to washout, and non-invasive techniques are required for reliable long-lasting recordings (*Guo et al., 2021a*; *Huson et al., 2023*). As such, it is very difficult to predict how the complete UBC population will respond to the complex activity patterns observed in vivo, given the complex dynamics from differential activation of AMPARs, mGluR1s and mGluR2/3s.

Here, we characterize UBC firing evoked by MF input patterns that are based on in vivo recordings of MF firing during smooth pursuit eye movements. We recorded MF-evoked increases in UBC spiking using cell-attached recordings in brain slices and pharmacologically assessed the contributions of mGluR2/3s, AMPARs, and mGluR1s. AMPARs conveyed time-locked responses to single stimuli, brief bursts, and baseline stimulation at up to 5 spikes/s (spk/s). But for bursts of 20 stimuli, excitatory responses are primarily mediated by mGluR1. An interplay of mGluR2/3s and mGluR1s dominated UBC responses to bursts of 20 stimuli or more and prolonged increases in input, creating a continuum of temporally-filtered responses. In this way a combination of glutamate receptors allows single MF to UBC connections to simultaneously convey a spike-timing component and a temporally filtered component.

# Results

## Diverse MF activity evokes wide ranging UBC responses

We set out to assess the UBC response to MF input patterns as observed in vivo in the vestibulocerebellum. For this purpose, we utilized in vivo recordings of MF activity during a smooth pursuit eye movement task from the floccular complex of rhesus macaques (provided by David J. Herzfeld and Stephen G. Lisberger) and reproduced the firing patterns observed there in acute cerebellar slices (*Figure 1a*). In their experiments, a head-fixed monkey was trained to track a smoothly moving visual target with minimal saccades while simultaneously recording single unit MF responses (see Methods). MF firing is comprised of sustained changes in firing that encode velocity and eye position, burst discharges that encode saccades, or a combination of both (*Lisberger and Fuchs, 1978*). We selected 30 s periods of firing from two representative MFs recorded in separate sessions. One showed characteristic burst firing (*Figure 1c*, green trace) and the other showed more sustained firing that increased or decreased in accordance with the smooth pursuit trials (*Figure 1d*, green trace). We performed cell-attached recordings from UBCs in acute cerebellar brain slices from P30-47 mice in the presence of GABA$_A$, GABA$_B$, and glycine receptor blockers (see Methods) and stimulated MF inputs using a theta-glass electrode.

As a means of categorizing the UBCs (*Guo et al., 2021a*; *Huson et al., 2023*), we stimulated the MF input with a standard burst of 20 stimuli at 100 spk/s (*Figure 1b*, green trace). These stimuli evoked diverse long-lasting changes in firing, with *Figure 1b* depicting examples of a fast UBC with relatively brief increases in firing, a longer lasting mid-range UBC, a slow UBC with a long-lasting increase in firing, and a UBC that is spontaneously active and transiently stops firing (OFF). We then applied stimulation patterns from in vivo recordings to these same UBCs. The first representative in vivo MF displayed a series of bursts consisting of 2–23 spikes at very high frequencies (*Figure 1c*, green trace) in line with previous findings (*Eccles et al., 1971*; *Rancz et al., 2007*). Activating MF inputs to the UBCs with this stimulation pattern evoked a series of responses that were qualitatively consistent with those evoked by 20 stimuli at 100 spk/s bursts. However, in the slow UBC, repeated bursts resulted in what appeared to be an increase in steady-state firing, and burst stimulation primarily induced a pause in this firing. Generally, different MF bursts evoked responses of different magnitude, which was particularly evident in the OFF cell, where even small changes in the input evoked noticeable decreases in firing.

Next, we stimulated MF inputs to the same four UBCs using an in vivo MF firing pattern associated with smooth pursuit trials (*Figure 1d*). This firing pattern consists of mossy fiber firing at 10–20 spk/s that periodically increases to 40–80 spk/s. The three excitatory UBCs responded with firing patterns that were qualitatively similar to the mossy fiber input, with irregular firing at 10–50 spk/s periodically increasing by 20–60 spk/s in response to elevations in mossy fiber firing. However, UBC responses were delayed relative to mossy fiber inputs, peaking later and remaining elevated after the MF firing rates decreased. As such, the firing of the excitatory UBCs appeared to be a low-pass filtered version of the MF input, with the time course progressively slower and the extent of filtering more pronounced for the fast, mid-range and slow cells. The slow cell increased its firing only after the elevation in MF firing. The OFF UBC completely shut down during the increase in MF firing, and only fired in between increases in firing, or in response to decreases in firing, effectively inverting the MF firing pattern. Overall, the different UBCs provided diverse temporal transformations of the in vivo MF input patterns.

## The influence of MF burst duration on UBC responses

Given the diversity of spike numbers in MF bursts in vivo, we systematically examined the effects of MF burst duration on the UBC response. Previously, we found that mossy fiber activation with a burst of 20 stimuli at 100 spk/s evoked a continuum of temporal responses in a population of UBCs (*Guo et al., 2021a*; *Huson et al., 2023*). We extended this approach by stimulating MFs with single stimuli, and 100 spk/s bursts of 2, 5, 10, and 20 stimuli (*Figure 2a*, green trace). We sorted UBCs (n=70) by their responses to the 20 stimuli at 100 spk/s burst, either according to the half-widths of their increases in firing (cells #1–61), or by the pause durations following stimulation (cells #62–70; *Guo et al., 2021a*; *Figure 2b*).

UBCs had highly diverse responses to different MF burst durations, as illustrated by four example cells (*Figure 2a*), and by the responses of all cells summarized in a heatmap (*Figure 2b*). Single

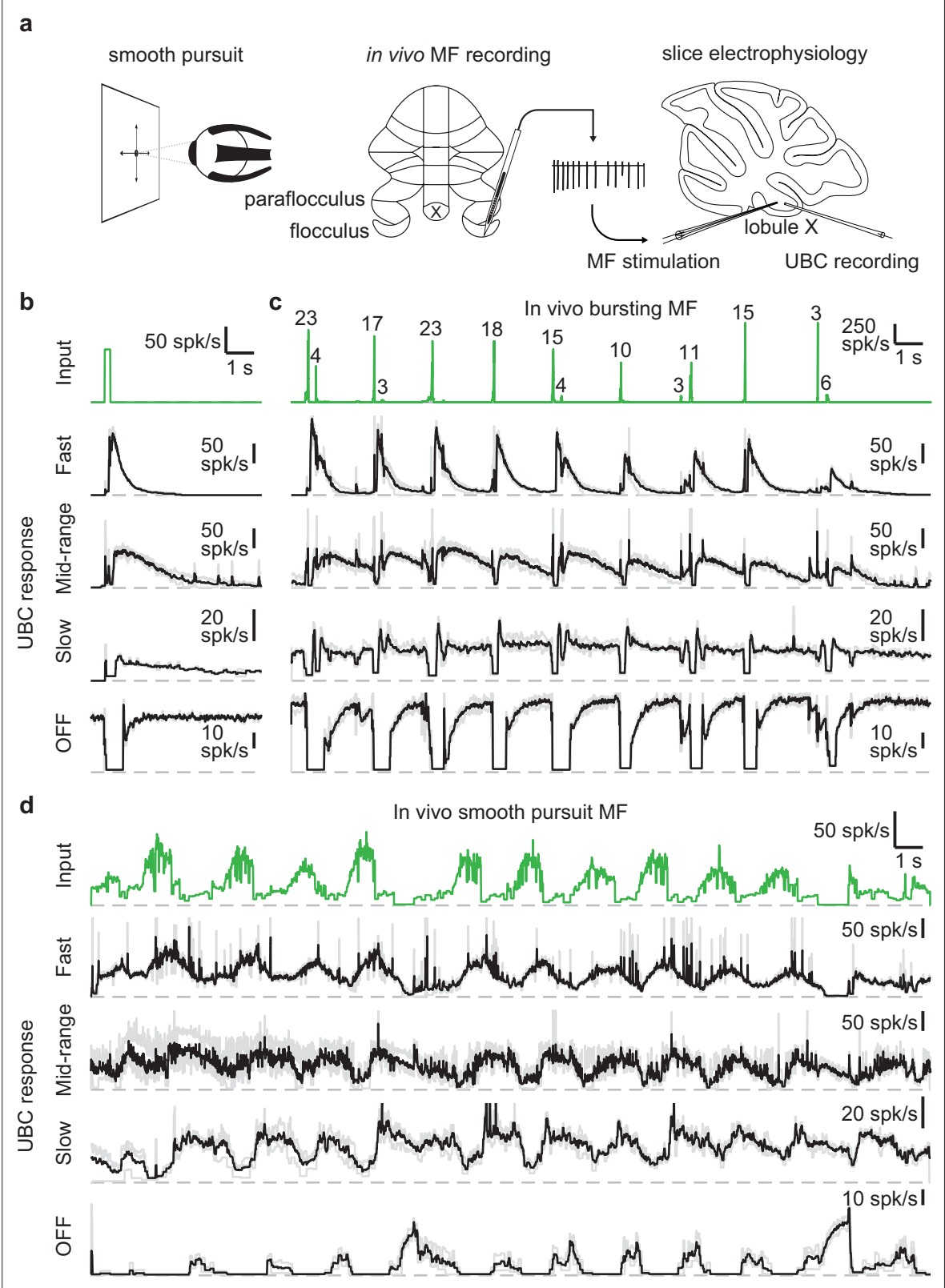

**Figure 1.** UBCs perform complex temporal transformations of in-vivo MF firing patterns. (**a**) Scheme showing experimental paradigm. We received in vivo recordings of MFs in the flocculus and paraflocculus of macaques (middle) during a smooth pursuit eye movement task (left). We reproduced the in vivo firing patterns in MFs in acute slices of lobule X of the mouse cerebellum using a theta stimulation electrode, while making cell-attached recordings of the UBC response (right). (**b**) MF stimulation with an artificial burst (20 stimuli at 100 spk/s; green trace, *top*) and instantaneous firing rates

*Figure 1 continued on next page*

*Figure 1 continued*

of the responses in 4 UBCs (fast, mid-range, slow, and OFF; individual trials in grey, mean in black). Dashed gray lines indicate 0 spk/s. (**c**) As in **b**, but for a MF firing pattern recorded in vivo during a smooth pursuit task in which the MF primarily fired in brief high frequency bursts. The numbers of spikes in each burst larger than 2 spikes are indicated in the trace. (**d**) As in **b**, but for a MF firing pattern recorded in vivo during a smooth pursuit task with characteristic long-lasting increases and decreases in firing.

stimuli and brief bursts evoked large rapid increases in firing in close to half of the UBCs (*Figure 2a*, cell 20; *Figure 2b*, most cells #1–40), and, surprisingly, peak firing rates were quite similar for single stimuli and 20 stimuli bursts for many of these UBCs. Different duration bursts evoked responses with very different time courses, as illustrated by cell #20 (*Figure 2a*), in which a single stimulus evoked a rapid increase in firing, but a 20-stimulus burst evoked a very short-lived increase at the onset of stimulation, before briefly stopping, and then resuming after stimulation ends. Single stimuli and brief bursts evoked decreases in firing for approximately 20% of the UBCs (*Figure 2a*, cells #55 and #67; *Figure 2b*, most cells #55–70). Single stimuli generally had very small effects on firing for UBCs with intermediate properties, but the size of the response increased as the stimulus duration increased (*Figure 2a*, cell #39; *Figure 2b*, most cells #41–54).

Summaries of peak increases in firing (*Figure 2c and d*), the number of spikes evoked by MF stimulation (*Figure 2e*), and the half-width of firing-rate increases (*Figure 2f*) for all burst durations revealed systematic trends in the properties of UBC responses. There were large differences in the dependence of peak firing on burst duration (*Figure 2c*). Fast UBCs responded to all burst durations with consistently high peak responses that exceeded 100 spk/s, while intermediate and slow UBCs increased their peak response almost linearly with burst duration (*Figure 2c*). As such, peak firing rates vary continuously across a wide range in response to 2 stimulus bursts, while in response to 20 stimulus bursts an apparent ceiling restricts the peak firing rates to a narrower range (*Figure 2d*). The number of spikes evoked by MF stimulation increased with burst duration in almost all cells, but increases were small in fast cells, and could grow more than 100-fold in slow cells (*Figure 2e*). The half-widths of responses also increased with input duration for almost all UBCs, with the largest increases apparent when the number of stimuli increases from one to five (*Figure 2f*).

Overall, these results indicate that UBCs can provide long-lasting responses to bursts with a wide range of input durations. After single stimuli and short bursts only fast UBCs respond with clear increases in firing, but these responses remain consistent as burst duration increases. This suggests that fast UBCs are suited to detecting the occurrence of a bust, regardless of burst duration. In contrast, slow UBCs are suited to detecting bursts of 5–20 MF spikes and scaling their responses proportionally to burst duration.

## Different glutamate receptors mediate distinct components of UBC responses

Many types of glutamate receptors are present at MF to UBC synapses, and it is not clear how each type contributes to responses evoked by MF bursts of different durations. We focused our studies on AMPARs and mGluR1s that elicit excitatory currents in UBCs, and mGluR2/3s that elicit inhibitory currents (*Rossi et al., 1995*; *Kinney et al., 1997*; *Russo et al., 2008*; *Kim et al., 2012*; *Schwartz et al., 2012*; *Borges-Merjane and Trussell, 2015*; *Guo et al., 2021a*). NMDARs are also present at UBC synapses, but we found that they do not make an appreciable contribution to responses evoked by bursts (*Figure 3—figure supplement 1*), and we therefore did not explicitly examine their contributions in combination with antagonists of the other glutamate receptors.

We evoked responses in 31 UBCs with single stimuli, 2, 5, 10, and 20 stimuli at 100 spk/s while additively washing in antagonists of mGluR2/3s (LY341495; 1–5 μM), AMPARs (NBQX; 5 μM), and mGluR1s (JNJ16259685; 1 μM) (*Figure 3a*, above). Blocking mGluR2/3 eliminated the pause after MF stimulation (e.g. cell #27, *Figure 3a*, orange, and *Figure 3d*, second row), in line with previous findings (*Russo et al., 2008*; *Kim et al., 2012*; *Borges-Merjane and Trussell, 2015*). Blocking mGluR2/3s also slightly increased peak responses (*Figure 3b*), and the number of evoked spikes in some cells (*Figure 3c and e*), especially for larger bursts, with significantly more spikes after blocking mGluR2/3s for burst of 5 stimuli or more (*Figure 3f*). The additional block of AMPARs eliminated increases in firing evoked by single stimuli in most cells (e.g. cell #8, *Figure 3a–c*, red, and *Figure 3d*, third row), but its effect diminished with increasing numbers of stimuli in the burst (*Figure 3b, c and e*), and on

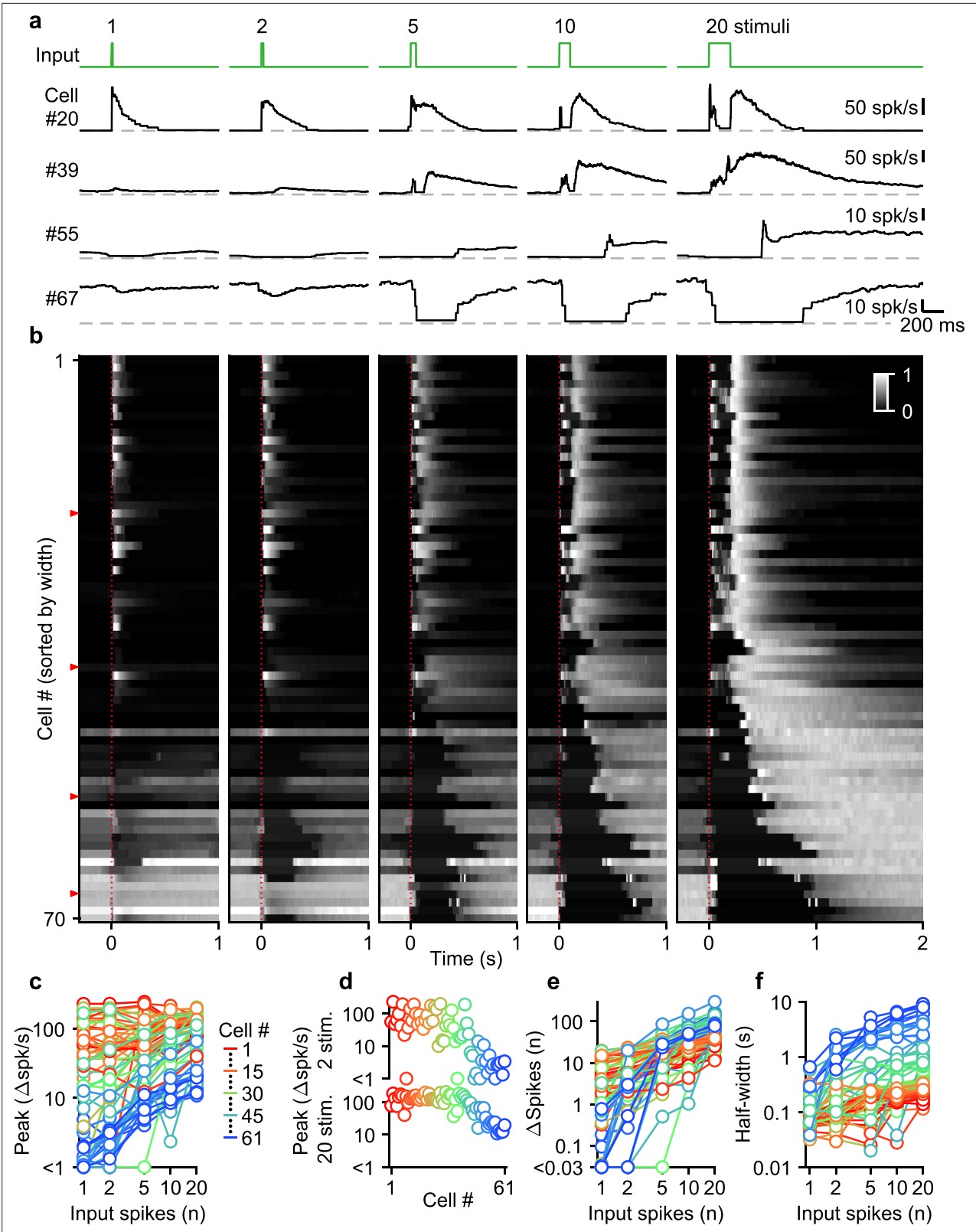

**Figure 2.** Diverse UBC responses to MF bursts of increasing duration. (**a**) Instantaneous firing rates of four representative UBCs in response to 100 spk/s bursts comprised of 1–20 stimuli (input pattern indicated above in green; dashed gray lines indicate 0 spk/s; cell numbers refer to the index in the summary plot **b**). (**b**) Heat maps showing the normalized instantaneous firing rates for all UBCs in response to bursts comprised of 1–20 stimuli. Responses were normalized per cell to the peak firing rate in response to 20 stimuli at 100 spk/s bursts, and sorted by the half-width for cells with clear

*Figure 2 continued on next page*

*Figure 2 continued*

increases in firing (cell #1–61), or sorted by pause duration for the rest (cell #62–70). Time indicates seconds since start of the MF stimulation (indicated by dotted red line). Red arrows indicate representative UBCs shown in **a**. (**c**) Summary plot showing peak increase in firing rate on a log scale, for all different bursts by the number of input spikes also on a log scale. UBCs are color coded to correspond to the cell index in **b**. Cells without significant increases in firing were excluded from this plot (#62–70). (**d**) Separate plots of the peak increase in firing rate in response to 2 and 20 stimuli bursts sorted by cell index as in **b**. (**e**) As in **c** but for the number of spikes evoked by the different MF burst stimulations. (**f**) As in **c** but for the half-width of the increase in firing for the different MF burst stimulations.

average AMPARs were not the main mediator of the increase in firing for bursts of 5 stimuli or more (*Figure 3g*). The component of the response that was sensitive to blocking mGluR1 increased as the bursts got longer and was the main mediator of the increase in firing in almost all UBCs for 20 stimulus bursts (*Figure 3b, c, e and h*). In a few UBCs, some short-lived responses remained after blocking both AMPARs and mGluR1s (e.g. cell #27 *Figure 3a and b*), but this did not represent a significant portion of the evoked spikes (*Figure 3c and e*).

AMPAR-mediated responses to single stimuli are more common in fast UBCs (*Figure 3d*). This is surprising since AMPAR subunits are expressed homogeneously across within the UBC population (*Guo et al., 2021a*; *Kozareva et al., 2021*; *Figure 3—figure supplement 2b*). It is therefore surprising that AMPAR responses are so small or are not apparent in slower UBCs and OFF UBCs. Although mGluR2/3 inhibition obscured small AMPAR components in some cells (e.g. cell #10, *Figure 3a*), this was uncommon and did not account for the differences in AMPAR-mediated responses in the UBC population (*Figure 3d*). It is possible that differential expression of auxiliary proteins governs the amount of AMPAR current present in a UBC. TARP γ–2 controls the amplitude of slow AMPAR EPSCs in UBCs (*Lu et al., 2017*), and it may be slightly elevated in UBCs that highly express mGluR1 (*Figure 3—figure supplement 2*), which corresponds to fast excitatory UBCs (*Borges-Merjane and Trussell, 2015*; *Guo et al., 2021a*; *Kozareva et al., 2021*). TARP γ–8 and TARP γ–7 have similar properties as TARP γ–2 (*Jackson and Nicoll, 2011*) and are also present in UBCs at low levels in UBCs. TARP γ–8 has an inverse expression pattern, and TARP γ–7 is expressed homogeneously, but their roles in regulating AMPARs in UBCs has not been examined. The expression of GSG1L, an AMPAR auxiliary subunit that is implicated in controlling AMPAR desensitization (*Shanks et al., 2012*), is of particular interest. GSG1L is expressed at high levels and shows a pronounced expression gradient (*Figure 3—figure supplement 2c*).

In conclusion, each type of glutamate receptor makes a distinct contribution to responses to bursts. mGluR2/3s mediate pauses in firing for both short and long bursts in over half of the cells. AMPARs are important mediators of firing rate increases evoked by short bursts in fast UBCs. mGluR1s are the primary mediators of responses to medium and long duration bursts.

## UBCs provide both time-locked and temporally-filtered responses

Next, we systematically characterized UBC responses evoked by stimulating MFs with input patterns that contain features that are characteristic of MF firing measured during smooth pursuit tasks. To achieve this, we exposed 31 cells to a smooth pursuit-like input pattern with prolonged baseline stimulation at 5 spk/s, interposed by 1 s steps to frequencies, ranging from 10 to 60 spk/s, at 4 s intervals (*Figure 4a*, green, top). We interposed this stimulation protocol with burst stimulations and used the responses to 20 stimulus at 100 spk/s bursts to sort cells for display purposes (*Figure 4a and c*, left).

UBCs showed diverse responses to this input pattern. For some UBCs, each stimulus during 5 spk/s baseline input evoked brief increases in firing, and steps to higher firing rates evoked slower, more prolonged increased in firing (*Figure 4*, cell #4). Other fast UBCs did not respond to 5 spk/s stimulation, but steps to 10–60 spk/s evoked responses that began with a delay and persisted after the step ended (*Figure 4*, cell #8). Along the sorted UBCs, 1 s steps evoked responses with increasingly delayed onset, and longer lasting increases in firing (*Figure 4*, cells #14 and #17). For slow cells, steps to 10–60 spk/s silenced firing during stimulation, and persistently elevated UBC firing after the step had ended (*Figure 4*, cells #22 and #27). Some slow UBCs also showed discrete decreases in firing during baseline 5 spk/s input (*Figure 4a and b*, cell #22), while others did not (*Figure 4a and b*, cell #27). These responses illustrate that UBCs provide both a temporal transformation of sustained changes in firing frequency, and discrete responses evoked by single stimuli. Single stimuli evoked

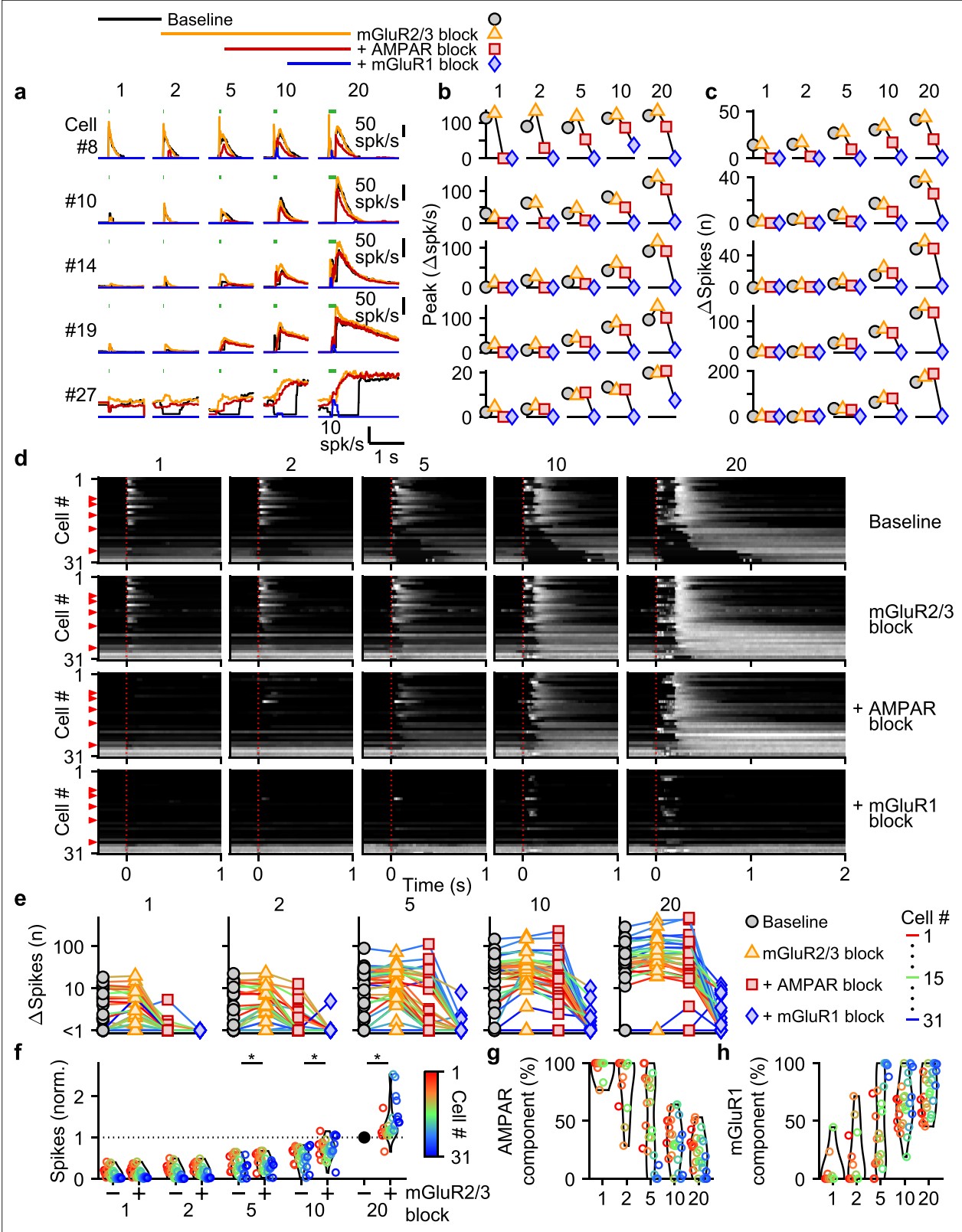

**Figure 3.** Differential glutamate receptor contributions to responses evoked by MF bursts of increasing duration. (**a**) Instantaneous firing rates of five representative UBCs in response to 100 spk/s bursts comprised of 1–20 stimuli (stimulus indicated by green bars). Responses shown for baseline conditions (black), and after addition of antagonists of mGluR2/3 (yellow), AMPAR (red), and mGluR1 (blue). Glutamate receptor antagonists were applied successively on top of the previous antagonist(s) as depicted in the scheme at the top. Cell numbers refer to the index in the summary plot

*Figure 3 continued on next page*

*Figure 3 continued*

(**d**). (**b**) Summaries of the peak change in firing rate for the same five cells, with successive different markers for baseline and the different glutamate antagonists, and separate plots for the different MF burst stimulations. (**c**) As in **b** but for the number of spikes evoked by the MF stimulation. (**d**) Heat maps showing the normalized instantaneous firing rates for all UBCs in response to 100 spk/s burst comprised of 1–20 stimuli (*columns*), for baseline and after addition of glutamate receptor antagonists (*rows*). Responses were normalized per cell to the peak firing rate in response to the 20 stimuli at 100 spk/s burst with mGluR2/3 blocked. Cells sorted by their response to the baseline 20 stimuli at 100 spk/s input, either by the half-width of the increase in firing (cell #1–27) or by pause duration (cell #28–31). Time indicates seconds since start of MF stimulation (indicated by dotted red line). Red arrows indicate representative UBCs shown in **a-c**. (**e**) Summary plots of the number of spikes evoked by the MF stimulation on a log scale for all UBCs color coded to correspond to the cell index in d. Successive different markers indicate baseline and the different glutamate antagonists, and separate plots for the different MF burst stimulations. (**f**) Violin plots of the number of spikes after each burst under baseline conditions (-) and after blocking mGluR2/3 (+), normalized per cell to the number of spikes after 20 stimuli at 100 spk/s under baseline conditions. Markers indicate individual UBCs color coded by the cell index in d. (*$P<0.01$, Wilcoxon signed rank test). (**g**) Violin plots of the percentage of the number of spikes evoked by MF stimulation that was mediated by AMPARs, estimated from the effect of blocking AMPARs on the response. Markers indicate individual UBCs color coded by the cell index in **d**. Responses smaller than 5 spikes not shown. (**h**) As in **g** but for the component mediated by mGluR1.

The online version of this article includes the following figure supplement(s) for figure 3:

**Figure supplement 1.** NMDA receptors do not significantly contribute to burst responses.

**Figure supplement 2.** AMPAR auxiliary subunits are expressed differentially in the UBC population.

changes in firing in a large percentage of cells for 5 spk/s stimulation, and responses were even larger for 1 spk/s and 2.5 spk/s (*Figure 4—figure supplement 1*).

As the step change of the input rate increased, the UBC population showed an increasingly wider range of responses. The number of evoked spikes that occurred during the 1 s steps increased for fast and intermediate UBCs, while slower UBCs tended to reduce their firing; with 45% of UBCs (14/31) reducing firing during the 60 spk/s step (*Figure 4e*). Step changes more consistently evoked increases in firing in the 3 s after the step, with 87% (27/31) of UBCs firing more spikes after a 60 spk/s step (*Figure 4f*), leaving only the slowest cells (#28–31) with net decreases in spiking. Some UBCs showed much larger responses for higher stimulus frequencies (e.g. cell #8 and #27, *Figure 4a*), while others were strikingly consistent (e.g. cell #4 and #17, *Figure 4a*). However, this property did not vary in alignment with the sorting of the UBCs by their half-width (*Figure 4e and f*). In cells with clear increases in firing after step changes (cells #1–27), we estimated time to peak for the responses to 60 spk/s steps by fitting the instantaneous firing rates with a log-gaussian function (see Methods). We observed gradually more delayed peaks in line with the UBC sorting ranging from less than 20ms to almost 2 s (*Figure 4g*). Half-decay times of the step responses were also highly correlated with the half-widths of the responses to 20 stimuli at 100 spk/s bursts (for the 60 spk/s step: Spearman's Rho = 0.873, p=$1.40 \times 10^{-6}$), and did not consistently change with firing frequency. Of the cells that showed increases in firing, 33% (9/27) took more than 1 s after the steps for their response amplitude to decay by half (*Figure 4h*), causing increases in firing to gradually build during the protocol, such that the response to step to 20 spk/s late in the protocol was far larger than one early in the protocol (e.g. cell #22 and #27, *Figure 4a*).

Overall, during smooth pursuit-like MF input, fast UBCs closely follow the input pattern, while slow UBCs produce strongly temporally filtered responses, in line with responses to 20 stimuli at 100 spk/s bursts. However, many UBCs additionally preserve timing information with discrete changes in firing time-locked to lower rate baseline stimuli. As such, multiple streams of information appear to be simultaneously encoded by the response.

## mGluR1 is the main mediator of UBC responses to smooth pursuit-like input patterns

We used selective antagonists to determine the roles of different glutamate receptors in mediating responses evoked by the stimulus protocol in *Figure 4a*. NMDARs do not make an appreciable contribution to responses evoked by this stimulus (*Figure 5—figure supplement 1*), and we therefore did not explicitly examine their contributions in combination with other glutamate-receptor antagonists. We again additively washed in antagonists of mGluR2/3s, AMPARs, and mGluR1s. For a fast UBC that responded to each 5 spk/s stimulation and to 1 s increases in input rate, blocking mGluR2/3s had minor effects, while blocking AMPARs eliminated responses to single stimuli only, and blocking mGluR1 eliminated responses to 1 s steps to 10–60 spk/s (*Figure 5a and b*, cell #3). In another cell

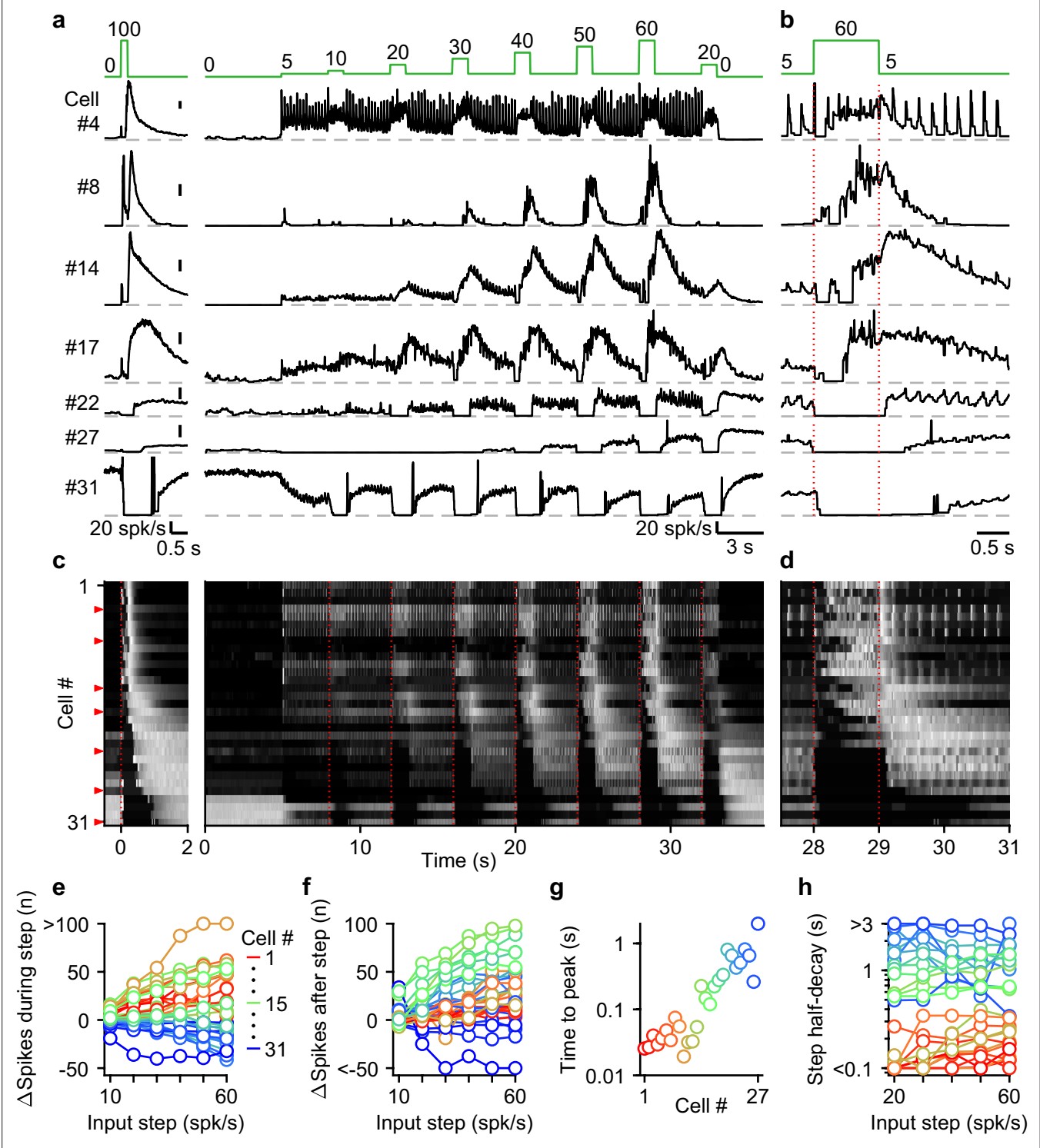

**Figure 4.** The UBC population displays a continuum of long-lasting responses to smooth pursuit-like input. (**a**) Instantaneous firing rates of seven representative UBCs in response to 20 stimuli at 100 spk/s burst (*left*), and smooth pursuit-like MF input (*right*) with prolonged 5 spk/s input interposed by 1 s step to 10–60 spk/s. Input pattern indicated above (green traces). Dashed gray lines indicate 0 spk/s; cell numbers refer to the index in the summary plot (**c**). (**b**) The same as in **a** but on an expanded timescale, displaying only the 60 spk/s step. Dotted red lines indicate onset and offset of the 60 spk/s step. (**c**) Heat maps showing the normalized instantaneous firing rates for all UBCs in response to 20 stimuli at 100 spk/s burst (*left*) and smooth pursuit-like MF input (*right*). Responses normalized to the peak firing rate per cell separately for burst and smooth pursuit-like MF input. Cell sorted by their response to 20 stimuli at 100 spk/s burst input, either by the half-width of the increase in firing (cell #1–25) or by pause duration (cell #26–31).

*Figure 4 continued on next page*

*Figure 4 continued*

Dotted red lines indicate the start of the step changes in input rate, red arrows indicate representative UBCs shown in **a,b**. (**d**) The same as in **c** but on an expanded timescale, displaying only the 60 spk/s step. (**e**) Summary plot of the number of evoked spikes during the 1 s steps compared to the 1 s period preceding the step. Individual UBCs color coded to correspond to the cell index in **c**. (**f**) As in **e** but for the number of evoked spikes in the 3 s period after the 1 s steps. (**g**) Summary plot of the time to peak for log-gaussian fits of the response to the step to 60 spk/s. Cells sorted and color coded to correspond to the cell index in **c**. Cells without significant increase in firing after step changes excluded (cell #28–31). (**h**) Summary plot of the time for the increase in firing to decay by half after a step change. Cells without significant increase in firing after step changes excluded (cell #28–31).

The online version of this article includes the following figure supplement(s) for figure 4:

**Figure supplement 1.** UBC responses to sustained 1–5 spk/s MF input.

(*Figure 5a and b*, cell #7), 5 spk/s stimulation did not evoke prominent responses, but step increases in input frequency evoked delayed and long-lasting increases in firing. In this cell, blocking mGluR2/3s shortened the delay between the step increases in MF stimulation and UBC firing rate increases (*Figure 5a and b*, cell #7), blocking AMPARs had very little effect, and blocking mGluR1s eliminated UBC responses. In slow UBCs, suppression of mGluR2/3s also removed discrete inhibitory responses to low-rate baseline input (*Figure 5—figure supplement 2b, d, f-h*). Blocking AMPARs eliminated excitatory responses evoked by individual stimuli at 5 spk/s (*Figure 5a–d*), 1 spk/s, and 2.5 spk/s stimuli (*Figure 5—figure supplement 2*).

For most UBCs, 1 s 10–60 spk/s MF stimulation evoked slow changes in UBC firing that remained prominent in the presence of mGluR2/3 and AMPAR antagonists. These responses were eliminated by blocking mGluR1 (*Figure 5a–d*). Plotting the change in spikes evoked by each 1 s step for the successive application of antagonists revealed that many UBCs showed clear increases in their excitatory response after suppression of mGluR2/3s (*Figure 5e and f*), which were also evident in the heatmaps (*Figure 5c and d*). Suppression of AMPARs only mildly reduced the number of evoked spikes for most steps (*Figure 5g*), with 90% (9/10) of UBCs that showed increases in firing to the 60 spk/s step before suppression of AMPAR, still responding after (*Figure 5e*). However, the number of evoked spikes was close to zero for all UBCs after suppression of mGluR1 (*Figure 5e and h*).

We conclude that, during smooth pursuit-like input patterns, mGluR1 is the main mediator of excitatory responses, while mGluR2/3 is the main mediator of inhibitory responses. and, additionally, reduces excitation overall throughout stimulation. For low-rate inputs in fast UBCs, AMPARs mediate an additional excitatory response that preserves timing information of the input, while mGluR2/3s provide an analogous inhibitory response in slower UBCs.

## Discussion

Our findings establish that the UBC population powerfully preprocesses MF inputs during realistic MF activation. UBCs simultaneously convey timing information, and a slow filtered version of the MF firing pattern that continuously varies across the population (*Figure 6b*). The multiple types of glutamate receptors present at MF-UBC synapses underlie the ability of UBCs to create diverse, multicomponent temporal responses in granule cells (*Figure 6c–e*).

### The roles of AMPARs in UBC responses to MF input

We find that for many UBCs AMPARs covey timing information about MF firing by responding to single stimuli, short bursts, and individual stimuli during prolonged low-frequency MF activation (*Figure 3*, *Figure 5*, *Figure 5—figure supplement 2*, *Figure 6c*). AMPARs primarily mediate responses to high frequency bursts of up to 5 stimuli, but as the burst duration increases their contribution diminishes and mGluR1s are the primary driver of increases in firing for most UBCs (*Figure 3*). A large fraction of the UBC population fire in response to each stimulus during 1–5 spk/s MF stimulation, and these responses are driven almost entirely by AMPARs (*Figure 4—figure supplement 1*, *Figure 5—figure supplement 2*). Although brief compared to UBC responses mediated by mGluR1s following prolonged bursts, AMPAR responses still last up to 400ms, which is much longer than AMPA-mediated EPSCs observed at most synapses. This slow time course is consistent with the properties of AMPAR EPSCs at MF-UBC synapses that arise from the ultrastructure of UBC synapses, the resulting slow glutamate signal, and the distinctive properties of the AMPARs present in UBCs (*Rossi et al., 1995*; *Kinney et al., 1997*; *van Dorp and De Zeeuw, 2014*; *Lu et al., 2017*; *Balmer et al., 2021*). Several

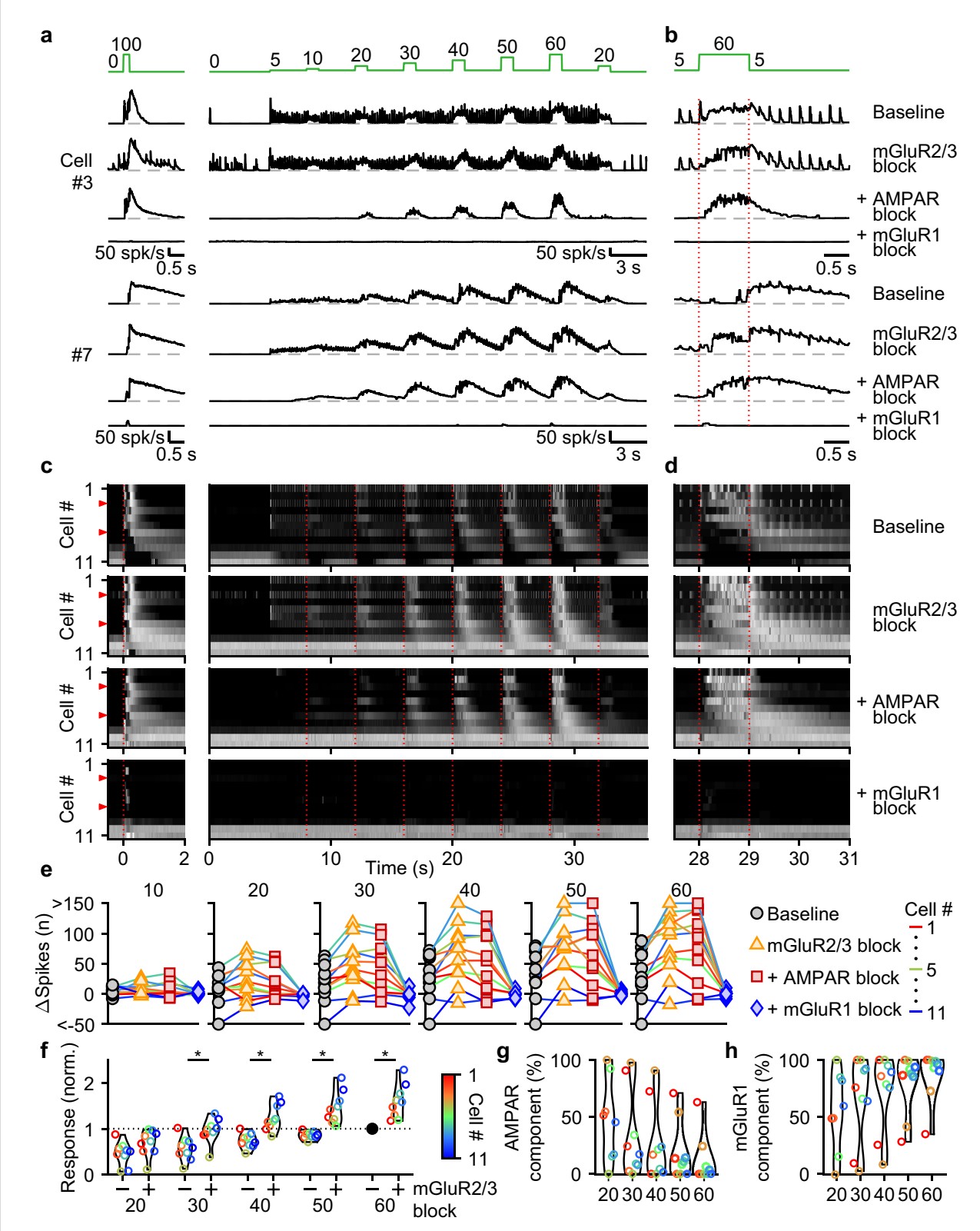

**Figure 5.** Contributions of mGluR2/3, AMPAR, and mGluR1 to responses evoked by smooth pursuit-like input. (**a**) Instantaneous firing rates of two representative UBCs in response to 20 stimuli at 100 spk/s bursts (*left*), and smooth pursuit-like MF input (*right*) as indicated by the traces at the top (green). Responses shown under baseline conditions and after successive addition of antagonists of mGluR2/3, AMPAR, and mGluR1. Dashed gray lines indicate 0 spk/s; cell numbers refer to the index in the summary plot (**c**). (**b**) The same as in **a** but on an expanded timescale, displaying only the 60 spk/s

*Figure 5 continued on next page*

*Figure 5 continued*

step. Dotted red lines indicate onset and offset of the 60 spk/s step. (**c**) Heat maps showing the normalized instantaneous firing rates for all UBCs in response 20 stimuli at 100 spk/s bursts (*left*) and smooth pursuit-like MF input (*right*), for baseline and after addition of glutamate receptor antagonists (*rows*). Responses normalized per cell separately for burst and smooth pursuit-like MF input to the respective peak firing rates with mGluR2/3 blocked. Cells sorted by their response to the baseline 20 stimuli at 100 spk/s input, either by the half-width of the increase in firing (cell #1–9) or by the pause duration (cell #10–11). Dotted red lines indicate the start of the step changes in input rate, red arrows indicate representative UBCs shown in **a**,**b**. (**d**) The same as in **c** but on an expanded timescale, displaying only the 60 spk/s step. (**e**) Summary plots of the number of evoked spikes in the 1 s period during the step and the 3 s period follow it for all UBCs color coded to correspond to the cell index in **c**. Successive different markers indicate baseline and the different glutamate antagonists, and separate plots for the different step changes in input. (**f**) Violin plots of the number of spikes in the 1 s period during the step and the 3 s period after under baseline conditions (-) and after blocking mGluR2/3 (+), normalized per cell to the number of spikes associated with the step to 60 spk/s under baseline conditions. Markers indicate individual UBCs color coded by the cell index in **c**. (*p<0.01, Wilcoxon signed rank test). (**g**) Violin plots of the percentage of the number of spikes evoked by MF stimulation that was mediated by AMPARs, estimated from the effect of blocking AMPARs on the response. Markers indicate individual UBCs color coded by the cell index in **d**. Responses smaller than 5 spikes not shown. (**h**) As in **g** but for the component mediated by mGluR1.

The online version of this article includes the following figure supplement(s) for figure 5:

**Figure supplement 1.** NMDA receptors do not significantly contribute to responses to smooth pursuit-like input.

**Figure supplement 2.** Contribution of different glutamate receptors to UBC responses evoked by sustained 1–5 Hz MF stimulation.

aspects of AMPAR-mediated spiking are readily explained by known properties of UBC synapses. The multiple phases of the AMPAR response evoked by 20 stimuli at 100 spk/s bursts likely reflect initial activation, desensitization during stimulation, and a large slow rebound current after stimulation as the prolonged presence of glutamate activates AMPARs as they recover from desensitization (*Figure 3a* cell #8, *Figure 3d*; *Kinney et al., 1997*; *van Dorp and De Zeeuw, 2014*; *Lu et al., 2017*; *Balmer et al., 2021*).The frequency-dependent decrease in AMPAR mediated burst magnitude (compare 1 and 5 spk/s *Figure 4—figure supplement 1*, *Figure 5—figure supplement 2*) also likely arises from AMPAR desensitization (*van Dorp and De Zeeuw, 2014*).

Contributions of AMPARs to MF-evoked responses are most apparent in fast UBCs (*Figure 2*), even though AMPAR subunits are expressed uniformly across the population (*Guo et al., 2021a*; *Kozareva et al., 2021*; *Figure 3—figure supplement 2b*). It is therefore surprising that AMPAR responses are so small or are not apparent in slower UBCs and OFF UBCs. It is possible that differential expression of auxiliary proteins such as TARPs or GSG1L governs the amplitude of AMPAR responses (*Figure 3—figure supplement 2c*), but further experiments are required to determine if this is the case.

## The roles of mGluR2/3s in MF-UBC responses

Suppression provided by mGluR2/3 receptors influences UBC responses in multiple ways (*Figure 6e*). (1) It suppresses baseline firing after a MF burst in 'OFF' UBCs, a role that has been extensively described previously (*Knoflach and Kemp, 1998*; *Russo et al., 2008*; *Kim et al., 2012*; *Borges-Merjane and Trussell, 2015*). (2) In most cells with a prominent pause in firing, suppression is followed by slow excitation mediated by mGluR1s. In these cells mGluR2/3s introduces a delay between the onset of MF stimulation and the response in the UBC. This is important for many UBCs that respond to 1 s MF activation with a delayed peak in firing that occurs long after the increase in MF firing rate. In this way mGluR2/3s crucially contribute to UBCs filtering of MF inputs by selectively influencing the slow rise of UBC spiking without influencing the time course of decay. (3) There is also an interplay between mGluR2/3 suppression and excitation by AMPARs and mGluR1s such that even in cells when suppression is not apparent, mGluR2/3s reduce the number of spikes evoked by stimulation. (4) In many UBCs, single stimuli lead to decreases that are mediated by mGluR2/3s.

## The roles of mGluR1s in MF-UBC responses

mGluR1s mediate slow responses that gradually build during repetitive activation (*Figure 6d*). Single stimuli and brief bursts build up too little glutamate to effectively activate mGluR1 and drive excitation (*Batchelor et al., 1994*; *Tempia et al., 1998*; *Brasnjo and Otis, 2001*; *Wadiche and Jahr, 2005*). mGluR1-mediated responses increase as the total number of spikes in the burst increases. In UBCs with prominent AMPAR responses, increasing the number of spikes in a high frequency burst leads to a transition from responses being primarily mediated by AMPARs to responses being mediated by mGluR1s. In these UBCs a combination of AMPARs and mGluR1s conveys relatively consistent

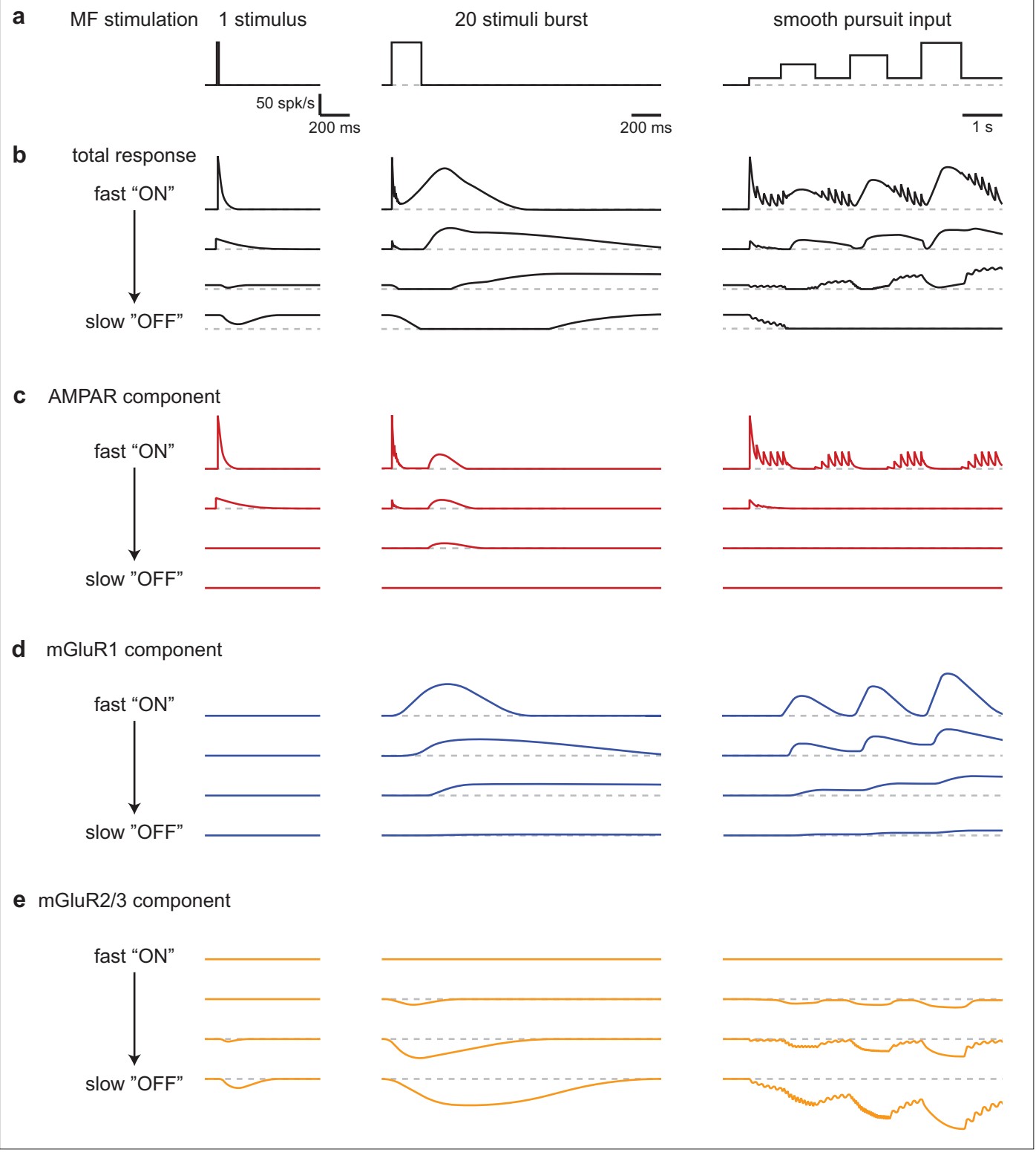

**Figure 6.** Schematic summarizing the range of UBC responses evoked by different stimuli. (**a**) MF stimuli used to evoke responses in UBCs. (**b–e**) Drawn representations of instantaneous firing rates illustrate the different types of UBC responses and the contributions of different types glutamate receptors. Four 'cells' are shown that range from fast 'ON' to slow 'OFF' cells that have the properties of classic ON and OFF cells, respectively. Two additional 'cells' with intermediate properties are also shown. (**b**) Total responses evoked by the stimuli in **a**. (**c**) Contributions of the AMPAR component to responses in **b**. (**d**) Contributions of the mGluR1 component to responses in **b**. (**e**) Contributions of the mGluR2/3 component to responses in **b**.

increases in peak firing that could not be achieved by either receptor alone (*Figures 2 and 3*). Such UBCs are highly effective at detecting the presence of a burst, but they are not suited to differentiating between bursts comprised of different numbers of MF spikes. In UBCs with a small AMPAR component and a prominent mGluR1 component, the UBC peak firing rate of the UBC response is strongly related to the number of spikes in the MF input burst. Similarly, increasing the firing rate of 1 s steps during prolonged MF input, as occurs during smooth pursuit eye movements, leads to increases in the magnitude of UBC responses. Additionally, mGluR1s also participate in temporally filtering MF input by mediating broad UBC responses of different durations (*Guo et al., 2021a*; *Huson et al., 2023*). However, during prolonged stimulation mGluR1s only play a crucial role in regulating the duration of UBC spiking following changes in the input rate, whereas the delay in the firing rate increase reflects the time course of mGluR2/3 suppression and is not limited by the time course of mGluR1 activation.

## Functional roles of MF transformations by the UBC population

Although the manner in which UBCs contribute to cerebellar-dependent behaviors has not been established, the properties of transformations at MF-UBC synapses provide insight into several possible functions. With regard to eye movements, the cerebellum is thought to aid in adaptation for real-time error correction during target tracking, and in learning to predict target trajectory both for saccades and smooth eye movements (*Shadmehr, 2017*; *Soetedjo et al., 2019*; *Lisberger, 2021*).

Saccadic eye movements are represented in MF signaling by rapid bursts preceding the onset of movement. The peak firing rate of a MF burst can encode the saccade amplitude (*Ohtsuka and Noda, 1992*). In Purkinje cells, eye velocity during saccades is encoded in the average response of the population, with individual cells showing diverse firing patterns including increases and decreases that can outlast the saccade (*Herzfeld et al., 2015*). Changes in the simple spike firing rate has been suggested to underly adaptation of saccadic eye movements (*Soetedjo et al., 2019*). UBC responses similarly consist of increases or decreases in firing that outlast the input, and as such, may contribute to the diversity in Purkinje cell firing rates, possibly providing a wider distribution of signals to support adaptation.

Visual feedback to the cerebellum lags eye movement by more than 100ms (*Raymond and Lisberger, 1998*). Therefore, to accurately track a target along its trajectory as during smooth pursuit, predictive eye movement signals are required. Temporal transformations by UBCs may be crucial to generating diverse signals to provide Purkinje cells with the appropriate substrate for learning, similar to how UBCs contribute to the diversity required for generating predictions of electrosensory feedback in weakly electric fish (*Kennedy et al., 2014*). Furthermore, mGluR2/3-generated delays in UBC responses are suited to encode timing, which is consistent with the hypothesis that timing rather than target position predicts the trajectory (*Medina et al., 2005*). Generally, learning a target trajectory takes place over hundreds of trials (*Medina et al., 2005*; *Hall et al., 2018*), but short-term plasticity is also exhibited when an instructive signal given in one trial biases eye movement in the next trial (*Yang and Lisberger, 2010*). This bias persists for 4–10 s, which is comparable to the durations of mGluR1-mediated increases in firing in slow UBCs (*Guo et al., 2021a*; *Huson et al., 2023*; *Figure 2b*). This raises the possibility that UBCs may provide a short-term plasticity signal that could be used to bias motion.

MF signals encoding vestibular modulation are also likely transformed by UBCs to enable behavioral adaptation. Previous work has highlighted that during sinusoidal vestibular input MF signals mostly remain in-phase, while granule cells fire in a range of different phases (*Arenz et al., 2008*; *Barmack and Yakhnitsa, 2008*). UBCs have been shown to phase shift sinusoidal inputs, enabling the cerebellum to learn to output signals at arbitrary shifts from the input which may enable learning of phase-shifted VOR (*Zampini et al., 2016*). That study focused on AMPAR and mGluR2/3 mediated changes in firing. The combination of delayed temporally-filtered responses mediated by mGluR2/3 and mGluR1, and the discrete changes in firing in response to single stimuli we describe here, enables the UBC population to provide transformations that are more complex than simple phase-shifts (*Figure 4*), and may be used for adaptations at longer time scales and to irregular input patterns.

Having separate response components also broadens the types of input UBCs can respond to, with AMPAR and mGluR2/3 responding to brief inputs, and mGluR1 and mGluR2/3 responding to long-lasting inputs. While UBCs are enriched in regions processing eye and vestibular signals, they are consistent circuit elements throughout the cerebellum (*Floris et al., 1994*; *Diño et al., 1999*; *Takács*

*et al., 1999*; *Englund et al., 2006*). The separate response components may provide UBCs with the flexibility required to usefully transform MF inputs in a wide range of behaviors.

## Possible Influence of Inhibition

Our experimental approach only evaluates the direct effects of MF activation on UBC firing and does not incorporate inhibition, which is expected to influence UBC firing in vivo. UBCs are inhibited by Golgi cells (*Rousseau et al., 2012*), and a subset of UBCs are also inhibited by PCs (*Guo et al., 2021a*). Although the potential influence of inhibition is of considerable interest, brain slice experiments are poorly suited to addressing this issue. Golgi cells are fragile and many do not survive in adult brain slices (*Hull and Regehr, 2012*), and many of the complex PC axon collaterals are severed before reaching UBCs (*Guo et al., 2021b*). In addition, it is exceedingly difficult to activate the population of PCs and Golgi cells as would occur in vivo. In contrast to the simplicity of a single MF input exciting a UBC, Golgi cells are directly activated by many MFs, by MF → granule cell → Golgi cell synapses, and by MF→UBC→ granule cell→Golgi cell synapses (*Hull and Regehr, 2022*), and PCs are activated by a great many MF → granule cell → PC synapses (*Witter et al., 2016*), by MF→UBC→ granule cell→PC→synapses and granule cell→molecular layer interneuron synapses (*Hull and Regehr, 2022*). As a result of these complexities, it will therefore be necessary to perform experiments in vivo to determine how Golgi cell and PC inhibition influences UBC transformations of MF inputs.

## Methods

### Key resources table

| Reagent type (species) or resource | Designation | Source or reference | Identifiers | Additional information |
|---|---|---|---|---|
| Strain, strain background (*Mus musculus*) | C57BL/6 | Charles River | RRID:IMSR_CRL:027 | |
| Chemical compound, drug | NBQX disodium salt | Abcam | Ab120046 | |
| Chemical compound, drug | (R)-CPP | Abcam | Ab120159 | |
| Chemical compound, drug | Strychnine hydrochloride | Abcam | Ab120416 | |
| Chemical compound, drug | Picrotoxin | Tocris | Cat. No. 1128 | |
| Chemical compound, drug | JNJ 16259685 | Tocris | Cat. No. 2333 | |
| Chemical compound, drug | LY 341495 | Tocris | Cat. No. 1209 | |
| Chemical compound, drug | CGP 55845 hydrochloride | Tocris | Cat. No. 1248 | |
| Software, algorithm | Igor Pro 8 | Wavemetrics; https://www.wavemetrics.com/ | RRID:SCR_000325 | |
| Software, algorithm | MafPC | Courtesy of MA Xu-Friedman; https://www.xufriedman.org/mafpc | | |
| Software, algorithm | MATLAB (R2023b) | MathWorks; https://www.mathworks.com/products/matlab.html | RRID:SCR_001622 | |

### Mice

All animal procedures were carried out in accordance with the NIH and Animal Care and Use Committee (IACUC) guidelines and protocols approved by the Harvard Medical Area Standing Committee on Animals (protocol #IS00000124). C57BL/6 mice of either sex from Charles River Laboratories were used for all experiments.

### Slice preparation

Juvenile (P30–P47) C57BL/6 mice of either sex were anesthetized by a peritoneal injection of 10 mg/kg ketamine/xylazine mixture and then transcardially perfused with ice-cold cutting solution (in mM): 110 choline chloride, 2.5 KCl, 1.25 $NaH_2PO_4$, 25 $NaHCO_3$, 25 glucose, 0.5 $CaCl_2$, 7 $MgCl_2$, 3.1 sodium pyruvate, and 11.6 sodium ascorbate, equilibrated with 95% $O_2$ and 5% $CO_2$. The brain was subsequently extracted, dissected, and submerged in the same solution. 220 µm parasagittal slices from the cerebellar vermis were cut on a Leica VT1200S vibratome. Slices were then incubated for 30 min at 34 °C in artificial cerebral spinal fluid (ACSF) containing (in mM): 125 NaCl, 26 $NaHCO_3$, 1.25 $NaH_2PO_4$,

2.5 KCl, 1 MgCl$_2$, 1.5 CaCl$_2$ and 25 glucose, equilibrated with 95% O$_2$ and 5% CO$_2$. Following incubation, the slices were kept for up to 6 hr at room temperature.

## Electrophysiology

Recordings were performed at ~34 °C in ACSF (flow rate 2–4 ml / min) containing inhibitory receptor blockers picrotoxin (20 µM; Tocris), CGP 55845 (1µM; Abcam), and strychnine hydrochloride (1 µM; Abcam). Visually guided cell-attached recordings were made in lobule X of the cerebellar vermis with 3–5 MΩ patch pipettes pulled from borosilicate glass capillaries (BF150-86-10, Sutter Instrument) with a P-97 Flaming/Brown puller (Sutter Instrument), and filled with ACSF. Once a loose seal was achieved (>10 MΩ), a bipolar theta glass pipette (BT150-10, Sutter Instrument) was placed in the granular layer more than 50 µm from the patched cell, and repositioned until a mossy fiber input was stimulated. UBCs were identified based on their slightly larger soma size compared to granule cells, and their distinctive long-lasting firing upon mossy fiber stimulation. Data were collected with a Multiclamp 700B amplifier (Molecular Devices), filtered at 4 kHz (4-pole Bessel filter) online, digitized at 20 kHz with an ITC18 (Heka Instrument), and saved using mafPC3 (custom software written by M. Xu-Friedman, https://www.xufriedman.org/mafpc) in Igor Pro 8 (WaveMetrics Inc) for offline analysis.

Our previous study (*Guo et al., 2021b*) explored issues related to the reliability of MF activation, the possibility of glutamate spillover from other synapses, and the possibility of disynaptic activation involving stimulation of MF→UBC→UBC connections (*Hariani et al., 2024*). We did on-cell recordings and followed that up with whole cell voltage clamp recordings from the same cell and there was good agreement with the amplitude and timing of spiking and the time course and amplitudes of the synaptic currents (*Guo et al., 2021b*). We also compared responses evoked by focal glutamate uncaging over the brush and MF stimulation and found that the time courses and amplitudes of the responses were remarkably similar (*Guo et al., 2021b*). This strongly suggests that the responses we observe do not reflect MF→UBC→UBC connections. We also showed that the responses were all-or-none: at low intensities no response was evoked, as the intensity of extracellular stimulation was increased a large response was suddenly evoked at a threshold intensity and further increases in intensity did not increase the amplitude of the response (*Guo et al., 2021b*).

## In vivo mossy fiber firing patterns

In vivo mossy fiber recordings were provided by David J. Herzfeld and Stephen G. Lisberger. Recordings were made using either single tungsten microelectrodes (FHC,~1 MΩ) or 16-channel Plexon S-probes in the ventral paraflocculus of a head-fixed rhesus monkey during a smooth pursuit eye movement task. Complete experimental details have been described previously (*Herzfeld et al., 2023*). Spikes were sorted using Full-binary pursuit (*Hall et al., 2021*), and manually curated by David J. Herzfeld. The mossy fiber traces reproduced in slice in this work were continuous 30 s recordings while the monkey was engaging in smooth pursuit eye movement trials. Briefly, smooth pursuit eye movement trials required the monkey to track a dot as it moves across a monitor at a constant speed. In each trial, the dot would appear stationary in the center of the monitor for 400–800ms before moving at a constant speed in one of 4 cardinal directions for 650ms. The dot would then remain stationary in an eccentric position for another 200ms.

## Pharmacology

Sequential drug wash-ins were performed by switching the bath solution to the same solution with addition of the antagonist(s), taking care to keep the flow rate consistent throughout (2 ml / min). Wash-ins were only performed when UBCs showed stable responses for at least 7.5 minutes. Cells with significant shifts in firing independent of mossy fiber stimulation during wash-ins were excluded from analysis. mGluR2/3s were blocked using LY 341495 (1–5 µM, Tocris). AMPARs were blocked using NBQX (5 µM, Abcam). mGluR1s were blocked using JNJ 16259685 (1 µM, Tocris). NMDARs were blocked using R-CPP (5 µM, Abcam).

## Quantification and statistical analysis

UBC responses were characterized using instantaneous firing rates in response to 20 stimuli at 100 spk/s bursts averaged over 5–10 median filtered trials recorded before other stimulation paradigms. Baseline firing was estimated from this using the second preceding mossy fiber stimulation. Pause

duration was calculated as the time between the end of the stimulation and the moment firing reached 5 spk/s or half the peak firing rate (whichever was smaller). The peak change in firing was calculated by subtracting the baseline firing rate from the peak firing rate after the end of MF stimulation. The half-width was calculated as the time between when the firing rate first exceeded and then decayed to half the peak amplitude (baseline corrected). UBCs were sorted by their half-width when the peak change in firing exceeded 8 spk/s (cell 1–61) or by their pause duration (cell 62–70).

Response parameters for different burst durations were calculated in a similar manner, with separate 20 stimuli at 100 spk/s trials recorded in proximity to other burst durations used for comparison. Additionally, the number of spikes evoked by MF stimulation was calculated by integrating the period from the start of stimulation till 2 times the UBCs half-width after the start of the response. The number of spikes was corrected for baseline firing. For smooth pursuit-like input patterns the number of spikes evoked by step changes in input were calculated in a similar manner but corrected for the average firing rate in the 1 s preceding the step change. To determine time to peak, responses to step changes were fit with a log-normal function, $A \cdot exp\left\{-\left[\left(ln\left(t\right)-\mu\right)/\sigma\right]^2\right\}$, where $A$ is the amplitude, $\mu$ is the peak location, and $\sigma$ is the width in log time. The half-decay of the response to step changes was calculated by finding the first point where the firing rate decreased to half the peak response, using an interpolated trace where responses to individual stimuli had been removed. For experiments with constant input at 1, 2.5, and 5 spk/s peak firing rates and number of spikes were calculated based on the average of all responses excluding the first 5, without correcting for baseline firing. For wash-in experiments all parameters were estimated based on trials 2–2.5 min after the start of the application of the antagonist, when responses to 20 stimuli at 100 spk/s bursts had reached a steady state. Significant differences in *Figures 3f and 5f* were determined using non-parametric Wilcoxon signed rank tests. Correlation was determined using Spearman's correlation coefficient. All analyses were done using MATLAB R2023b (MathWorks).

## Acknowledgements

We thank Stephen G Lisberger and David J Herzfeld for providing in vivo mossy fiber recordings and for their comments and input on the project. We thank members of the Regehr lab for comments on the manuscript. The work was supported by the NIH R35NS097284.

## Additional information

### Funding

| Funder | Grant reference number | Author |
|---|---|---|
| National Institute of Neurological Disorders and Stroke | R35NS097284 | Wade G Regehr |

The funders had no role in study design, data collection and interpretation, or the decision to submit the work for publication.

### Author contributions

Vincent Huson, Conceptualization, Data curation, Formal analysis, Investigation, Visualization, Methodology, Writing – original draft, Writing – review and editing; Wade G Regehr, Conceptualization, Resources, Supervision, Funding acquisition, Writing – original draft, Project administration, Writing – review and editing

### Author ORCIDs

Vincent Huson http://orcid.org/0000-0002-3556-1436
Wade G Regehr https://orcid.org/0000-0002-3485-8094

### Ethics

All animal procedures were carried out in accordance with the NIH and Animal Care and UseCommittee (IACUC) guidelines and protocols approved by the Harvard Medical Area StandingCommittee on Animals (protocol #IS00000124).

Reviewer #1 (Public review): https://doi.org/10.7554/eLife.102618.3.sa1
Reviewer #2 (Public review): https://doi.org/10.7554/eLife.102618.3.sa2
Author response https://doi.org/10.7554/eLife.102618.3.sa3

## Additional files

### Supplementary files
MDAR checklist

### Data availability
All data that support the findings in this study have been deposited in Harvard Dataverse at https://doi.org/10.7910/DVN/EBL6C3.

The following dataset was generated:

| Author(s) | Year | Dataset title | Dataset URL | Database and Identifier |
|---|---|---|---|---|
| Huson V, Regehr WG | 2024 | Unipolar brush cell responses to realistic mossy fiber input | https://doi.org/10.7910/DVN/EBL6C3 | Harvard Dataverse, 10.7910/DVN/EBL6C3 |

The following previously published dataset was used:

| Author(s) | Year | Dataset title | Dataset URL | Database and Identifier |
|---|---|---|---|---|
| Kozareva V, Martin C, Osorno T, Rudolph S, Guo C, Vanderburg C, Nadaf N, Regev A, Regehr W, Macosko E | 2021 | A transcriptomic atlas of mouse cerebellar cortex reveals novel cell types | https://singlecell.broadinstitute.org/single_cell/study/SCP795/a-transcriptomic-atlas-of-the-mouse-cerebellum | Single Cell Portal, SCP795 |

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
