## [Editor Report · eLife Assessment]

This **valuable** study describes how trains of mossy fiber stimulation control cerebellar unipolar brush cell discharges. The dissection of the contributions of relevant glutamate receptors to these transformations is **convincing**. Overall, the study broadens our understanding of temporal processing in the cerebellar cortex.

---

## [Referee Report · Reviewer #1 (Public review)]

In this manuscript, the authors recorded cerebellar unipolar brush cells (UBCs) in acute brain slices. They confirmed that mossy fiber (MF) inputs generate a continuum of UBC responses. Using systematic and physiological trains of MF electrical stimulation, they demonstrated that MF inputs either increased or decreased UBC firing rates (UBC ON vs. OFF) or induced complex, long-lasting modulation of their discharges. The MF influence on UBC firing was directly associated with a specific combination of metabotropic glutamate receptors, mGluR2/3 (inhibitory) and mGluR1 (excitatory). Ultimately, the amount and ratio of these two receptors controlled the time course of the effect, yielding specific temporal transformations such as phase shifts. The experiments are well-executed and properly analyzed.

Strengths:

(1) A wide range of MF stimulation based on activity patterns observed in vivo was explored, including burst duration and frequency dependency, which could serve as a valuable foundation for explicit modeling of temporal transformations in the granule cell layer.

(2) The pharmacological blockade of mGluR2/3, mGluR1, AMPA, and NMDA receptors helped identify the specific roles of these glutamate receptors.

(3) The experiments convincingly demonstrate the key role of mGluR1 receptors in temporal information processing by UBCs.

Weaknesses:

(1) This study is a follow up of previous work (Guo et al., Nat. Commun., 2021).

(2) The MF activity used to mimic natural stimulation was previously collected from primates, whereas the recordings were conducted in mice.

Comments on revisions:

The authors included a discussion about inhibition, but I still disagree with their claim that it was not possible to study the MF-UBC connection with inhibition unblocked. This group has already conducted experiments on Golgi cell inhibition in slices.

---

## [Referee Report · Reviewer #2 (Public review)]

This study addresses the question of how UBCs transform synaptic input patterns into spiking output patterns and how different glutamate receptors contribute to their transformations. The first figure utilizes recorded patterns of mossy fiber firing during eye movements in the flocculus of rhesus monkeys obtained from another laboratory. In the first figure, these patterns are used to stimulate mossy fibers in the mouse cerebellum during extracellular recordings of UBCs in acute mouse brain slices. The remaining experiments stimulate mossy fiber inputs at different rates or burst durations, which is described as 'mossy-fiber like', although they are quite simpler than those recorded in vivo. As expected from previous work, AMPA mediates the fast responses, and mGluR1 and mGluR2/3 mediate the majority of longer-duration and delayed responses. The manuscript is well organized and the discussion contextualizes the results effectively.

Comments on revisions:

The authors have adequately addressed my concerns.

---

## [Author Response]

The following is the authors’ response to the original reviews.

**Public Reviews:**

**Reviewer #1 (Public review):**
In this manuscript, the authors recorded cerebellar unipolar brush cells (UBCs) in acute brain slices. They confirmed that mossy fiber (MF) inputs generate a continuum of UBC responses. Using systematic and physiological trains of MF electrical stimulation, they demonstrated that MF inputs either increased or decreased UBC firing rates (UBC ON vs. OFF) or induced complex, long-lasting modulation of their discharges. The MF influence on UBC firing was directly associated with a specific combination of metabotropic glutamate receptors, mGluR2/3 (inhibitory) and mGluR1 (excitatory). Ultimately, the amount and ratio of these two receptors controlled the time course of the effect, yielding specific temporal transformations such as phase shifts.Overall, the topic is compelling, as it broadens our understanding of temporal processing in the cerebellar cortex. The experiments are well-executed and properly analyzed.Strengths:(1) A wide range of MF stimulation patterns was explored, including burst duration and frequency dependency, which could serve as a valuable foundation for explicit modeling of temporal transformations in the granule cell layer.(2) The pharmacological blockade of mGluR2/3, mGluR1, AMPA, and NMDA receptors helped identify the specific roles of these glutamate receptors.(3) The experiments convincingly demonstrate the key role of mGluR1 receptors in temporal information processing by UBCs.Weaknesses:(1) This study is largely descriptive and represents only a modest incremental advance from the previous work (Guo et al., Nat. Commun., 2021).

We feel that the present study is a major advance. It builds on (Guo et al., Nat. Commun., 2021) in which we examined the effects of bursts of 20 stimuli at 100 spk/s. In that study we found that differential expression of mGluR1 and mGluR2 let to a continuum of temporal responses in UBCs, but AMPARs make a minimal contribution for such bursts. It was not known how UBCs transform realistic mossy fiber input patterns. Here we provide a comprehensive evaluation of a wide range of input patterns that include a range of bursts comprised of 1-20 stimuli, sustained stimulation with stimulation of 1 spk/s to 60 spk/s. This more thorough assessment of UBC transformations combined with a pharmacological assessment of the contributions of different glutamate receptor subtypes provided many new insights:

• We found that UBC transformations are comprised of two different components: a slow temporally filtered component controlled by an interplay of mGluR1 and mGluR2, and a second component mediated by AMPARs that can convey spike timing information. NMDARs do not make a major contribution to UBC firing. The finding that UBCs simultaneously convey two types of signals, a slow filtered response and responses to single stimuli, has important implications for the computational potential of UBCs and fundamentally changes the way we think about UBCs.

• We found that with regard to the slow filtered component mediated by mGluR1 and mGluR2, we could extend the concept of a continuum of responses evoked by 20 stimuli at 100 spk/s (Guo et al., Nat. Commun., 2021) to a wide range of stimuli. It was not a given that this would be the case.

• The contributions of AMPARs was surprising. Even though snRNAseq data did not reveal a gradient of AMPAR expression across the population of UBCs (Guo et al., Nat. Commun., 2021), we found that there was a gradient of AMPA-mediated responses, and that the AMPA component was also most prominent in cells with a large mGluR1 component. Our finding that AMPAR accessory proteins exhibit a gradient across the population, which could account for the gradient of AMPAR responses, will prompt additional studies to test their involvement.

(2) The MF activity used to mimic natural stimulation was previously collected in primates, while the recordings were conducted in mice.

Our first task was to determine the firing properties of mossy fibers under physiological conditions in UBC rich cerebellar regions. Previous studies have estimated this in anesthetized mice using whole cell granule cell recordings (Arenz et al., 2008; Witter & De Zeeuw 2015). However, for assessing firing patterns during awake behavior, we felt that the most comprehensive data set available in a UBC rich cerebellar region was for mossy fibers involved in smooth pursuit in monkeys (David J. Herzfeld and Stephen G. Lisberger). This revealed the general features of mossy fiber firing that helped us design stimulus patterns to thoroughly probe the properties of MF to UBC transformations. The firing patterns are designed to investigate the transformations for a wide range of activity patterns and have important general implications for UBC transformations that are likely applicable to UBCs in different species that are activated in different ways.

(3) Inhibition was blocked throughout the study, reducing its physiological relevance.

The reviewer correctly brings up the very important issue of inhibition in shaping UBC responses. It is well established that UBCs are inhibited by Golgi cells (Rousseau et al., 2012), and we recently showed that some UBCs are also inhibited by PCs (Guo et al., eLife, 2021). This will undoubtedly influence the firing of UBCs in vivo. We considered examining this issue, but felt that brain slice experiments are not well suited to this. In contrast to MF inputs that can be activated with a realistic activity pattern, it is exceedingly difficult to know how Golgi cells and Purkinje cells are activated under physiological conditions. Each UBC is activated by a single mossy fiber, but inhibition is provided by Golgi cells that are activated by many mossy fibers and granule cells, and PCs that are controlled by many granule cells and many other PCs. In addition, we found that many Golgi cells do not survive very well in slices, and the axons of many PCs are severed in brain slice. Although limitations of the slice preparation prevent us from determining the role of inhibition in shaping UBC responses, we have added a section to the discussion in which we address the important issue of inhibition and UBC responses.

**Reviewer #2 (Public review):**
This study addresses the question of how UBCs transform synaptic input patterns into spiking output patterns and how different glutamate receptors contribute to their transformations. The first figure utilizes recorded patterns of mossy fiber firing during eye movements in the flocculus of rhesus monkeys obtained from another laboratory. In the first figure, these patterns are used to stimulate mossy fibers in the mouse cerebellum during extracellular recordings of UBCs in acute mouse brain slices. The remaining experiments stimulate mossy fiber inputs at different rates or burst durations, which is described as 'mossy-fiber like', although they are quite simpler than those recorded in vivo. As expected from previous work, AMPA mediates the fast responses, and mGluR1 and mGluR2/3 mediate the majority of longer-duration and delayed responses. The manuscript is well organized and the discussion contextualizes the results effectively.The authors use extracellular recordings because the washout of intracellular molecules necessary for metabotropic signaling may occur during whole-cell recordings. These cell-attached recordings do not allow one to confirm that electrical stimulation produces a postsynaptic current on every stimulus. Moreover, it is not clear that the synaptic input is monosynaptic, as UBCs synapse on one another. This leaves open the possibility that delays in firing could be due to disynaptic stimulation. Additionally, the result that AMPAmediated responses were surprisingly small in many UBCs, despite apparent mRNA expression, suggests the possibility that spillover from other nearby synapses activated the higher affinity extrasynaptic mGluRs and that that main mossy fiber input to the UBC was not being stimulated. For these reasons, some whole-cell recordings (or perforated patch) would show that when stimulation is confirmed to be monosynaptic and reliable it can produce the same range of spiking responses seen extracellularly and that AMPA receptormediated currents are indeed small or absent in some UBCs.

We appreciate the reviewer’s concerns regarding the reliability of mossy fiber activation, the possibility of glutamate spillover from other synapses, and the possibility of disynaptic activation involving stimulation of MFàUBCàUBC connections. We examined these issues in a previous study (Guo et al., Nat. Commun., 2021). We did on-cell recordings and followed that up with whole cell voltage clamp recordings from the same cell (Guo et al., Nat. Commun., 2021, Fig. 5), and there was good agreement with the amplitude and timing of spiking and the time course and amplitudes of the synaptic currents. We also compared responses evoked by focal glutamate uncaging over the brush and MF stimulation (Guo et al., Nat. Commun., 2021, Fig. 4). We found that the time courses and amplitudes of the responses were remarkably similar. This strongly suggests that the responses we observe do not reflect disynaptic activation (MFàUBCàUBC connections). We also showed that the responses were all-or-none: at low intensities no response was evoked, as the intensity of extracellular stimulation was increased a large response was suddenly evoked at a threshold intensity and further increases in intensity did not increase the amplitude of the response (Guo et al., Nat. Commun., 2021, Extended data Fig. 1). We can be well above threshold and still excite the same response, and as a result we do not see stereotyped indications of an inability to stimulate during prolonged high frequency activation. We recognize the importance of these issues, so we have added a section dealing explicitly with these issues (pp. 15-16).

A discussion of whether the tested glutamate receptors affected the spontaneous firing rates of these cells would be informative as standing currents have been reported in UBCs. It is unclear whether the firing rate was normalized for each stimulation, each drug application, or each cell. It would also be informative to report whether UBCs characterized as responding with Fast, Mid-range, Slow, and OFF responses have different spontaneous firing rates or spontaneous firing patterns (regular vs irregular).

The spontaneous firing of UBCs is indeed an interesting issue that is deserving of further investigation. It is not currently known how spontaneous firing at rest is regulated in UBCs, however, in previous work we have shown that there is great diversity in the rates across the population of UBCs in the dorsal cochlear nucleus (Huson & Regehr, JNeurosci, 2023, Fig. 4). Unfortunately, during the kind of sustained high-frequency stimulation protocols (as used in this study) spontaneous firing rates tend to increase. This is likely an effect of residual receptor activation. As such, our current dataset is not suitable to performing in depth analysis of the effects of the different glutamate receptors on spontaneous firing rates. As this study aims to explore UBC responses to MF inputs we feel that specific experiments to address the issue of spontaneous firing rates are outside of the scope.

As the reviewers points out there are indeed different ways the firing rates can be normalized for display in the heatmaps, and different normalizations have been used in different figures. We have made sure that the method for normalization is clearly indicated in the figure legends for each of the heatmaps on display, specifying the protocol and drug application used for normalization.

Figure 1 shows examples of how Fast, Mid-range, Slow, and OFF UBCs respond to in vivo MF firing patterns, but lacks a summary of how the input is transformed across a population of UBCs. In panel d, it looks as if the phase of firing becomes more delayed across the examples from Fast to OFF UBCs. Quantifying this input/output relationship more thoroughly would strengthen these results.

The UBC responses to in vivo MF firing patterns are intriguing and we agree that there appears to be increasing delays for slower UBCs visible in Figure 1. However, we feel that the true in vivo MF firing patterns are too complex and irregular for rigorous interpretation. Therefore, we only tested simplified burst and smooth pursuit-like input patterns on the full population of UBCs. Here we indeed do see increasingly delayed responses as UBCs get slower (Fig. 4).

Inhibition was pharmacologically blocked in these studies. Golgi cells and other inhibitory interneurons likely contribute to how UBCs transform input signals. Speculation of how GABAergic and glycinergic synaptic inhibition may contribute additional context to help readers understand how a circuit with intact inhibition may behave.

As indicated in our response to reviewer 1, we have added a section discussing the very important issue of inhibition and UBC responses in vivo.

**Recommendations for the authors:**

**Reviewer #1 (Recommendations for the authors):**
(1) Including recordings without inhibition blocked would strengthen the study and provide a more comprehensive view of the transformations made by UBCs at the input stage of the cerebellar cortex.

See response to public comments.

(2) The authors claim that a continuum of temporal responses was observed in UBCs, but they also distinguish between fast, mid-range, slow, and OFF UBCs. While some UBCs fire spontaneously, others are activated by MF inputs. A more thorough classification effort would clarify the various response profiles observed under specific MF stimulation regimes. Have the authors considered using machine learning algorithms to aid in classification?

We fundamentally feel that these response properties do not conform to rigid categories. In our previous work we have shown that UBC population constitutes a continuum in terms of gene expression, and in terms of spontaneous and evoked firing patterns. While in order to answer some questions empirically it may still be useful to apply advanced algorithms to enforce separate groups to be compared, in this work we aimed to present the full range of UBC responses without introducing any additional biases that such methods would produce.

(3) A robust classification could assist in quantifying the temporal shifts observed during smooth pursuit-like MF stimulation, a critical outcome of the study.

As stated above, we prefer to present an unbiased overview of the continuous nature of the UBC population, as we believe that this is fundamentally the most accurate representation. While it is true that this prevents us from providing a quantification in the different temporal shifts, we believe that the range of shifts across the population is sufficiently large and continuously varying the be convincing (see Figure 4d).

(4) In Figure 5, contrary to what is described on page 10, Cells 10 and 11 (OFF UBCs) appear to behave differently, as mGluR1 does not seem to affect their firing rates. A specific case should be made for OFF UBCs.Indeed, cells 10 and 11 do not show clear increases in firing and are not strongly affected by blocking of mGluR1. However, as discussed above and explored in our previous work, we feel that the range of UBC increases in firing is best described as a continuum, including the extreme where increases in firing are no longer clearly observable. As the aim in this work is to describe this continuum of responses for physiologically relevant inputs, we do not feel there is a benefit to creating a specific case for OFF UBCs here. It should be pointed out that the number of “pure” OFF UBCs completely lacking an mGluR1 component is very small.(5) A summary diagram should be added at the end of the manuscript to highlight the key temporal features observed in this study.

This is a great suggestion and we have prepared such a summary diagram (Figure 6).

**Reviewer #2 (Recommendations for the authors):**
(1) Page 3- "Assed" should be "assessed"(2) Page 19- "by integrating" is repeated twice(3) It was not noted whether the data would be made available. It could be useful for those interested in implementing UBCs in models of the cerebellar cortex.

We agree that this data set is invaluable to those interested in implementing UBCs in models of the cerebellar cortex. We will make the dataset available as described in the text.